# A review of rate control in atrial fibrillation, and the rationale and protocol for the RATE-AF trial

Dipak Kotecha,[1,2,3,4] Melanie Calvert,[4,5] Jonathan J Deeks,[5,6] Michael Griffith,[2] Paulus Kirchhof,[1,2,3,4] Gregory YH Lip,[1,3,4] Samir Mehta,[6] Gemma Slinn,[6] Mary Stanbury,[7] Richard P Steeds,[1,2] Jonathan N Townend[1,2]

DK is the Chief Investigator

[1]Institute of Cardiovascular Sciences, University of Birmingham, Birmingham, UK
[2]Cardiology, University Hospitals Birmingham NHS Trust, Birmingham, UK
[3]Cardiology, Sandwell & West Birmingham Hospitals NHS Trust, Birmingham, UK
[4]Centre for Patient Reported Outcomes Research, University of Birmingham, Birmingham, UK
[5]Institute of Applied Health Research, University of Birmingham, Birmingham, UK
[6]Birmingham Clinical Trials Unit, University of Birmingham, Birmingham, UK
[7](Lead for the Patient and Public Involvement panel), Birmingham, UK

**Correspondence to**
Dr. Dipak Kotecha;
d.kotecha@bham.ac.uk

## ABSTRACT

**Background and objective** Atrial fibrillation (AF) is common and causes impaired quality of life, an increased risk of stroke and death as well as frequent hospital admissions. The majority of patients with AF require control of heart rate. In this article, we summarise the limited evidence from clinical trials that guides prescription, and present the rationale and protocol for a new randomised trial. As rate control has not yet been shown to reduce mortality, there is a clear need to compare the impact of therapy on quality of life, cardiac function and exercise capacity. Such a trial should concentrate on the long-term effects of treatment in the largest proportion of patients with AF, those with symptomatic permanent AF, with the aim of improving patient well-being.

**Design and intervention** The RAte control Therapy Evaluation in permanent Atrial Fibrillation (RATE-AF) trial will enrol 160 participants with a prospective, randomised, open-label, blinded end point design comparing initial rate control with digoxin or bisoprolol. This will be the first head-to-head randomised trial of digoxin and beta-blockers in AF.

**Participants** Recruited patients will be aged ≥60 years with permanent AF and symptoms of breathlessness (equivalent to New York Heart Association class II or above), with few exclusion criteria to maximise generalisability to routine clinical practice.

**Outcome measures** The primary outcome is patient-reported quality of life, with secondary outcomes including echocardiographic ventricular function, exercise capacity and biomarkers of cellular and clinical response. Follow-up will occur at 6 and 12 months, with feasibility components to inform the design of a future trial powered to detect a difference in hospital admission. The RATE-AF trial will underpin an integrated approach to management including biomarkers, functions and symptoms that will guide future research into optimal, personalised rate control in patients with AF.

**Ethics and dissemination** East Midlands-Derby Research Ethics Committee (16/EM/0178); peer-reviewed publications.

**Trial registration** Clinicaltrials.gov: NCT02391337; ISRCTN: 95259705. Pre-results.

## Strengths and limitations of this study

► Control of heart rate is universally used in patients with atrial fibrillation (AF), but evidence from good quality randomised controlled trials is extremely limited.

► Despite common clinical use, there has never been a direct randomised comparison of beta-blockers and digoxin for heart rate control in patients with AF (with or without heart failure).

► The RAte control Therapy Evaluation in permanent Atrial Fibrillation (RATE-AF) trial will assess the effect of therapy on patient-reported quality of life, and improve methods to capture this information in patients with AF. The trial will also evaluate the long-term impact on cardiac function, define reproducible methods to measure systolic and diastolic function in AF and develop new biomarkers for personalisation of treatment.

► The trial will not have the power to identify differences in clinical events, but will allow us to plan a future trial designed to detect a difference in the need for admissions to hospital.

## INTRODUCTION

Atrial fibrillation (AF) is a common cause of stroke and cardiovascular death, leads to poor quality of life and doubles the risk of hospital admission.[1] We are currently in the midst of an epidemic of AF, with both incidence and prevalence expected to double in the next 20 years.[2–4] Although AF can affect any age group, patients are typically elderly with significant comorbidities, including up to 50% suffering from heart failure.[5] AF is both a cause and consequence of heart failure, with complex interactions leading to impairment of systolic and diastolic function.[6 7] The combination of these two conditions is expected to have a dramatic impact on the burden of healthcare worldwide.[8–11]

Management of AF involves anticoagulation to prevent strokes, selecting appropriate patients for restoration of sinus rhythm and almost universal need for control of heart rate. In contrast to other management strategies, the choice of rate

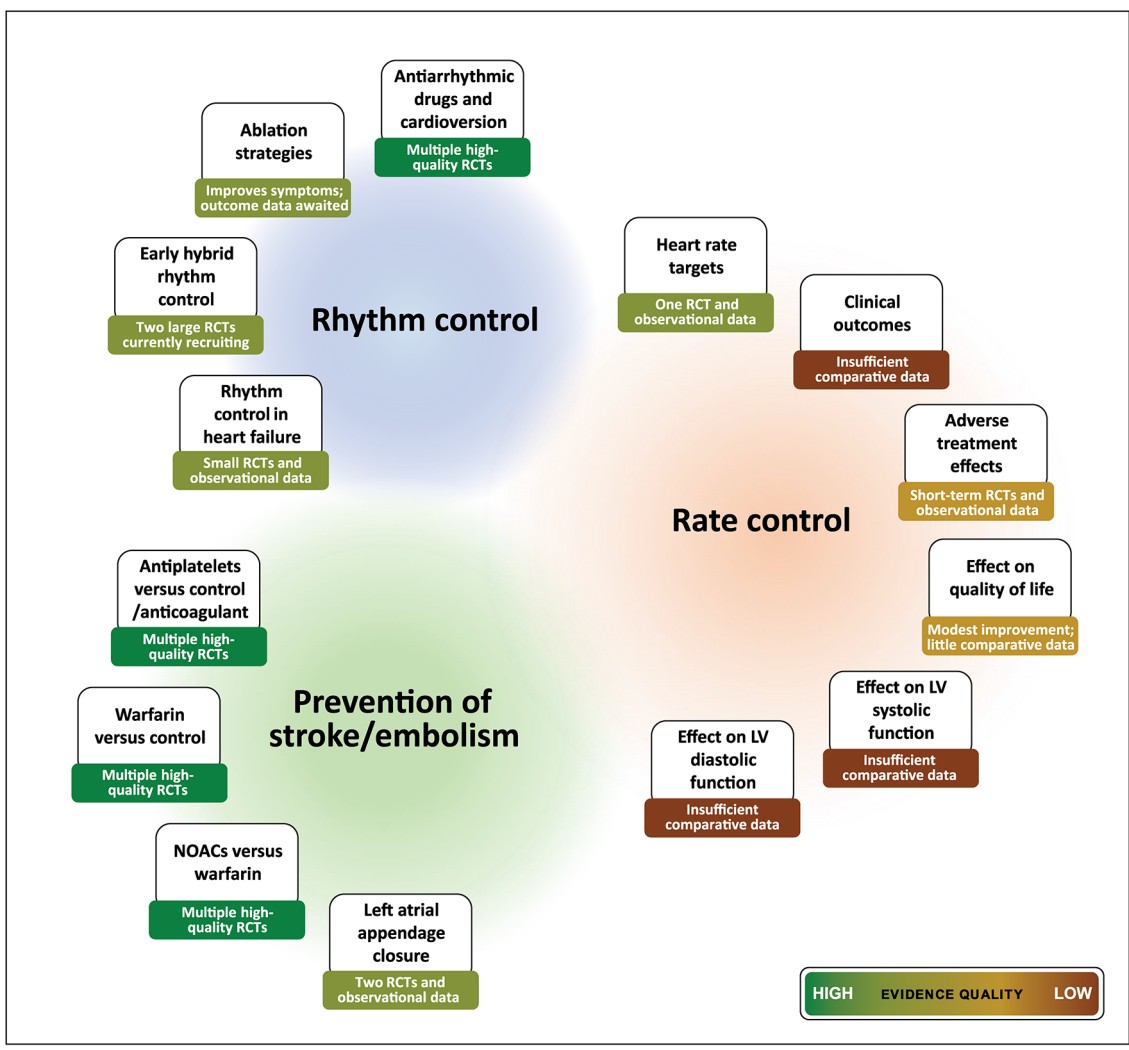

**Figure 1** Evidence-based summary for management of atrial fibrillation. Summary of evidence for main components of clinical management, highlighting paucity of robust data for key issues regarding rate control therapy. RCT, randomised controlled trial; LV, left ventricular; NOAC, novel oral anticoagulants.

control therapy has a very low-quality evidence base (figure 1).[12] Guidelines from the National Institute for Health and Care Excellence and the European Society of Cardiology (ESC) have mandated further research specifically on rate control,[1 13] which is also reflected in the level of recommendations from the American Heart Association.[14] The small studies currently available are often uncontrolled or with short follow-up,[15–19] providing few insights on the biological effects of treatment or the mechanisms underpinning the response to therapy. With no evidence for any impact of rate control on mortality,[20 21] and limited data for any difference in quality of life or functional outcomes, the choice of rate control agent is currently informed by expert consensus and physician experience.

In this paper, we review the current evidence-base for rate control in AF and the rationale for a new randomised controlled trial (RCT). The RAte control Therapy Evaluation in permanent Atrial Fibrillation (RATE-AF) trial will compare initial therapy with beta-blockers versus digoxin in older patients with symptomatic permanent

AF, assessing quality of life, functional capacity, left-ventricular ejection fraction (LVEF), diastolic function and biomarkers of treatment response.

## Rationale for a new trial of rate control in AF
### Why not choose a rhythm control strategy?
A number of RCTs have assessed the addition of rhythm control strategies to control of heart rate in patients with AF, most often with anti-arrhythmic drugs (AAD) and direct current cardioversion. Neither of the two largest trials found any difference in clinical outcomes comparing these approaches (Atrial Fibrillation Follow-up Investigation of Rhythm Management (AFFIRM) and Rate Control versus Electrical Cardioversion for Persistent Atrial Fibrillation (RACE)).[22 23] Other smaller trials and meta-analyses have confirmed that rhythm control is not superior to regulation of heart rate alone,[24–26] including heart failure patients with both impaired and preserved ejection fraction.[27 28] These studies have analysed heterogeneous populations, including both paroxysmal and permanent AF that may differ with regard to mechanism,

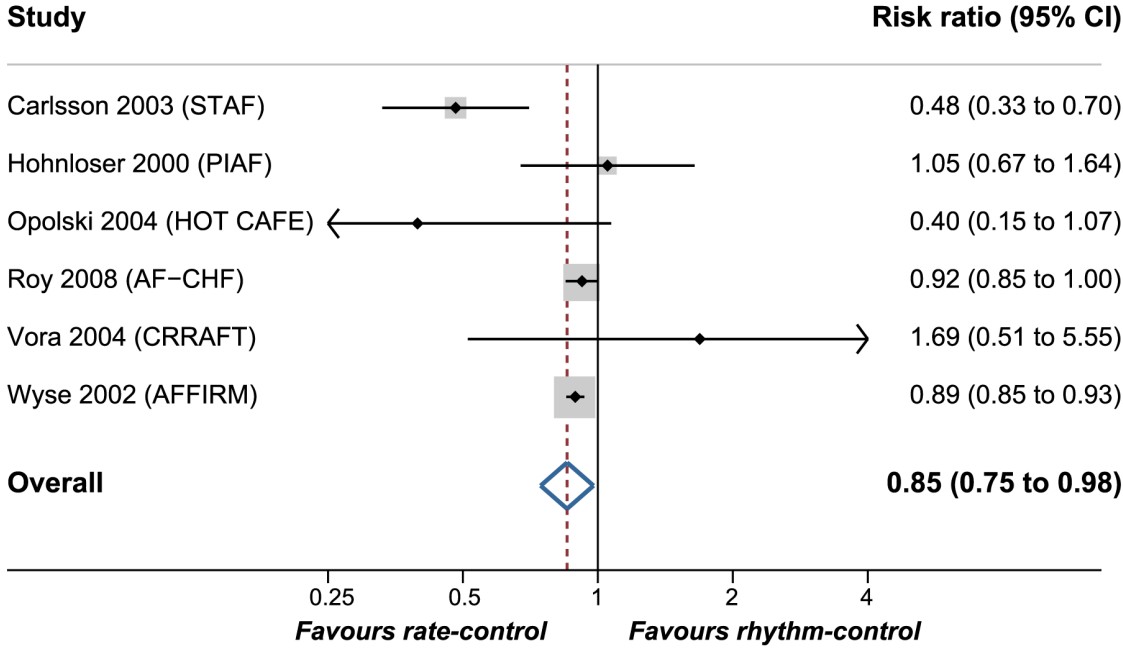

**Figure 2** Hospitalisation in rate vs rhythm control trials. Meta-analysis of hospitalisation in the six largest rate vs rhythm control trials, excluding hospital visits for cardioversion procedures, where applicable. Studies are pooled with a random-effects model. Significant heterogeneity was identified, with an $I^2$ value of 66.8% (p=0.01). Grey boxes represent the comparative weight of the study. STAF, Strategies of Treatment of Atrial Fibrillation study (cardioversion/AAD vs rate control in persistent AF)[76]; PIAF, Pharmacological Intervention in Atrial Fibrillation trial (amiodarone/cardioversion vs diltiazem in persistent AF)[77]; HOT CAFE, How to Treat Chronic Atrial Fibrillation study (cardioversion/AAD vs rate control in persistent AF)[78]; AF-CHF, Atrial Fibrillation and Congestive Heart Failure trial (cardioversion/AAD vs rate control in paroxysmal/persistent AF with LVEF ≤35%)[27]; CRAAFT, Control of Rate vs Rhythm in rheumatic Atrial Fibrillation Trial (cardioversion/amiodarone vs diltiazem in persistent AF due to rheumatic heart disease)[79]; AFFIRM, Atrial Fibrillation Follow-up Investigation of Rhythm Management study (AAD/cardioversion versus rate control in paroxysmal/persistent AF); AAD, anti-arrhythmic drugs; LVEF, left-ventricular ejection fraction.[22]

prognosis and the response to treatment.[15] However, there is also evidence that a rhythm control strategy may increase hospital admissions. A meta-analysis of major published trials is presented in figure 2, highlighting a 17% increase in the risk of hospitalisation in the rhythm control group (after exclusion of hospital visits related to cardioversion). Although limited by patient crossover and the association between AAD and adverse events,[29] the results highlight the importance of trials comparing different rate control options and associated healthcare costs.

Although AF ablation is becoming increasingly popular, it remains a highly invasive method to restore sinus rhythm.[30 31] Current guidelines recommend ablation to improve AF-related symptoms in patients with paroxysmal AF, or as a treatment option in symptomatic persistent AF that is refractory to other therapy.[1 14] Long-term outcome studies are awaited and need to be balanced against procedural complications and AF recurrence. Even in patients receiving intensive rhythm control therapy, rate control is often necessary to reduce symptoms during AF paroxysms. Furthermore, 40%–50% of patients with AF are deemed as unsuitable for rhythm control (permanent

AF),[5 32] and are maintained on rate control therapy to reduce potential symptoms and avoid tachycardia that may worsen ventricular function.[6] Patients with permanent AF have a higher residual risk of cardiovascular death, stroke or systemic embolism, despite anticoagulation.[33]

### What is the optimal heart rate target in AF?
There is clinical uncertainty about how to control heart rate and the intensity of rate-reduction. In the RACE II trial of 614 randomised patients with permanent AF, there were no benefits of strict (<80 bpm at rest) compared with lenient rate control (resting heart rate <110 bpm) over 3 years of follow-up.[34] Although interpretation was limited by the narrow difference in heart rate between groups, lenient rate control was found to be non-inferior with an adjusted HR of 0.80 (90% CI 0.55 to 1.17) for a wide composite of adverse clinical outcomes (12.9%, compared with 14.9% in the strict control arm). In addition, there were no differences in symptoms or New York Heart Association (NYHA) class,[34 35] and patients achieving strict rate control required more clinic visits.[36] These findings are consistent with other trials,[37–39] registries[32] and even

randomised[40] and observational[41] cohorts in patients with concomitant heart failure, suggesting that intensity of heart rate control is not the key determinant of outcomes in AF.

## Do outcomes vary with different rate control therapies?

Medical therapy to achieve rate control in AF can be achieved with beta-blockers, digoxin and non-dihydropiridine calcium channel blockers (CCB; diltiazem or verapamil).[1] Only a limited evidence-base is available to assist clinicians in choosing first-line and subsequent therapy, resulting in wide variations in clinical practice[42–44] and frequent use of combination therapy. Guidelines suggest the choice of medication should be individualised, dependent on the presence of ongoing symptoms.[1 14] However, these recommendations are based on low-quality trials and observational data, often with small numbers of participants and follow-up over a few weeks.[16] There are no RCTs comparing long-term rate control options in AF.

Demonstrating any reduction in hard clinical outcomes with rate control has proved elusive. In patients with heart failure, reduced ejection fraction and concomitant AF, an individual patient-level meta-analysis of double-blind RCT data has suggested that beta-blockers do not reduce all-cause mortality or hospital admissions compared with placebo,[20] in contrast to the substantial benefit seen in sinus rhythm.[45] Similarly, the use of digoxin was not associated with any increase, or reduction, in mortality in a comprehensive systematic review.[21] This finding deviates from prior observational analyses which are confounded by the fact that sicker patients tend to receive digoxin more often, which can only be addressed within a randomised trial. Although digoxin is known to reduce hospital admissions in patients with heart failure and reduced ejection fraction in sinus rhythm,[46] the impact in patients with AF is unknown.

If rate control has limited effect on mortality, what about evidence for a differential effect on other outcomes, such as functional capacity, cardiac function or quality of life? Beta-blockers are the most commonly used rate control agents and although they have a greater impact than digoxin on heart rate during exertion, there is no evidence that this results in better exercise capacity.[17 18 47–49] Beta-blockers were not associated with any improvement in arrhythmia-related symptoms in a small RCT of 60 low-risk patients with permanent AF, compared with diltiazem and verapamil which reduced the frequency of symptoms.[50] Those in the beta-blocker group had a reduction in exercise capacity and increase in B-type natriuretic peptide (BNP) compared with those treated with CCB.[51] Analysis of smaller trials comparing beta-blockers with CCB are inconsistent.[17] Compared with verapamil or diltiazem, digoxin has less effect on heart rate but there is no consistent evidence for any difference in functional outcomes.[17 18 47 49 52] Importantly, diltiazem and verapamil are usually avoided in patients with reduced ejection fraction due to the risk of adverse

outcomes,[53–57] leaving only beta-blockers or digoxin as suitable therapy. Only a single RCT has been published comparing beta-blockers with digoxin in patients with AF and heart failure (mean LVEF 24%, n=47).[58] Although there was a marginally significant improvement in LVEF with combined carvedilol/digoxin versus placebo/digoxin, blinded withdrawal of digoxin then led to a deterioration in LVEF, accompanied by an increase in BNP. The direct effects of digoxin on LVEF and diastolic function have only been studied in patients with sinus rhythm, where digoxin increased LVEF by 3%–11% and improved diastolic filling.[59–61] Magnesium has been shown to complement digoxin therapy to achieve lower ventricular rates in AF,[62] but is not in common use due to the availability of beta-blockers and CCB which are more potent agents for acute heart rate control.[1] Although data on patient-reported quality of life are limited,[63 64] rate control has been associated with improved quality of life in trials assessing rate versus rhythm control.[65–67] The mechanism by which rate control therapy mediates an increase in physical functioning and quality of life is unknown but conceivably due to improvements in LVEF and/or diastolic function.

In summary, rate control is an important part of treatment in all patients with AF but the evidence-base is poor, particularly in those with permanent AF who form the majority of patients in clinical practice. Rate control in AF is also subject to considerable, and poorly characterised individual variability in response, with limited information about the effects of therapy on cardiac function, quality of life and functional capacity.

## The RATE-AF trial

The RATE-AF trial is the first head-to-head randomised assessment of beta-blockers versus digoxin as the initial rate control agent in patients with AF. The trial has a prospective, randomised, open-label, investigator-blinded end point (PROBE) design, and is planned as an inclusive study that reflects and will have an important impact on clinical practice (box 1). The primary outcome is patient-reported quality of life using the Short Form (36) Health Survey (SF-36) physical component summary score at 6 months' post-randomisation. The major secondary outcomes are change in LVEF and diastolic function on echocardiography, functional capacity, global and AF-specific quality of life and cardiovascular biomarkers (box 2). A key objective of the trial is to improve the methods used for measuring quality of life in patients with AF, as well as optimising the validity, reproducibility and acquisition of echocardiographic left-ventricular function. The RATE-AF trial will also act as a feasibility study to plan a future, event-driven clinical trial exploring the impact of different rate control strategies on cardiovascular events and unplanned hospital admissions. The study is sponsored by the University of Birmingham and funded by the National Institute for Health Research (NIHR), as part of a Career Development Fellowship awarded to the Chief Investigator (DK).

**Box 1    The RAte control Therapy Evaluation in permanent Atrial Fibrillation (RATE-AF) trial—information for patients**

**About atrial fibrillation**
Atrial fibrillation is a common heart condition that leads to an irregular and often rapid heart rate. Atrial fibrillation causes 1 in 4 strokes, and patients have frequent hospital admissions and a higher risk of dying. In addition, atrial fibrillation makes many patients feel unwell, with reduced quality of life.

**What is the purpose of the trial?**
Atrial fibrillation usually requires medication to control heart rate, but we currently do not know which medication is better for patients. The aim of this study is to find out which of the two treatments improves quality of life and the function of the heart, digoxin or bisoprolol (a beta-blocker).

**What will happen in the trial?**
The RATE-AF trial is designed to compare two approaches for control of heart rate, based on initial treatment with either digoxin or beta-blockers, medications which are commonly used by doctors. The main objective of the trial is to research the effects of treatment on quality of life in patients with atrial fibrillation . We will also test whether quality of life questionnaires respond to changes in symptoms experienced by patients, how we use ultrasound to look at the function of the heart, and develop new markers in the blood to personalise treatment.

**More information**
RATE-AF trial video: https://www.youtube.com/watch?v=4oxe8AcVo0E or search 'rateaf' in YouTube.
Patient information (British Heart Foundation): https://www.bhf.org.uk/heart-health/conditions/atrial-fibrillation.

## METHODS
### Patients
Inclusion criteria are patients aged 60 years or older with breathlessness (equivalent to NYHA class II or more) and permanent AF, characterised as a physician decision for rate control with no plans for cardioversion, AAD or ablation therapy. Only limited exclusion criteria apply (figure 3), reflecting any clear requirements or contraindications for either beta-blockers or digoxin. As neither agent impacts on mortality in patients with heart failure,[20 21] reduced LVEF is not an exclusion criterion. All patients are expected to be anticoagulated if appropriate, according to their clinical risk of stroke and thromboembolism.

### Study procedures and outcomes
One hundred and sixty eligible patients in need of rate control will be invited to participate in the study from primary and secondary care across two major NHS Trusts in Birmingham, UK. The RATE-AF trial is managed by the Birmingham Clinical Trials Unit (University of Birmingham) and situated within the Birmingham NIHR/Wellcome Trust Clinical Research Facility.

Following written informed consent, participants will be randomised in a 1:1 ratio to either bisoprolol or digoxin therapy. Randomisation will be provided by a computer-generated minimisation algorithm to ensure balance between the treatment arms for baseline European Heart Rhythm Association class and gender. Allocation will be concealed until the patient has been recruited and consented, thereafter the trial will be open-label.

Baseline assessment procedures will include patient-reported quality of life questionnaires (table 1), 6 min walk distance, echocardiography and biomarker assessment. Participants will then receive study medication (bisoprolol 1.25–15 mg once daily or low-dose digoxin 62.5–250 µg once daily), with scheduled uptitration visits to attain a heart rate at rest of ≤100 bpm. This heart rate is in line with international recommendations[1] and was chosen pragmatically to reflect the opinion of many cardiologists that tachycardia can lead to, or worsen, systolic and diastolic dysfunction. Ambulatory 24 hours ECG monitoring will be performed at the end of uptitration (unblinded). Investigator-blinded end points will be assessed at the interim (6-month) and final (12-month) visit, which include patient-reported quality of life, echocardiographic parameters of systolic and diastolic left-ventricular function and biomarker assessment (figure 3).

### Exploratory work and clinical practice improvement
During the trial, qualitative research using focus groups and structured interviews will assess whether the quality of life questionnaires adequately and acceptably assess changes in symptom burden in a sample of patients from each treatment arm. We will also compare and contrast the generic and AF-specific questionnaires. The aim of this work is to improve the methods used for measuring patient-reported outcomes in AF, and to address some of the limitations we have identified in published validation studies.[68]

Optimal acquisition of echocardiography in patients with AF will be determined by reproducibility studies, comparing repeated measures of systolic/diastolic function according to cardiac cycle length. The RATE-AF trial will address the evidence gaps we have identified in a systematic review of echocardiography in patients with AF,[69] and explore the diagnostic difficulty of categorising heart failure in the context of AF (particularly with preserved ejection fraction, where symptom classification is confounded and BNP levels are consistently raised due to AF.[7]

Blood samples from participants will be analysed for the cellular effects of rate control, including intracellular sodium, calcium and endogenous cardiotonic steroids (CTS) using photometry in cultured human cardiomyocytes. This work will give mechanistic insight into the cellular response to beta-blockers and digoxin, identify novel markers of treatment effect and develop assays that are more robust than serum digoxin concentration (SDC) for determining individual patient dosage. SDC is an immunoassay known to be a poor marker of digoxin toxicity,[70] which can cross-react with other targets[71] (eg, endogenous CTS). Although SDC will be performed at 6 months follow-up and as required during the trial to advise clinicians on dose and avoid high digoxin levels, digoxin toxicity remains a clinical diagnosis at present.

Box 2   Outcomes and objectives of the RAte control Therapy Evaluation in permanent Atrial Fibrillation (RATE-AF trial

**Primary outcome:**
Comparison of two strategies for rate control on patient-reported quality of life, based on initial use of digoxin vs beta-blocker therapy, with a predefined focus on physical well-being using the SF-36 physical component summary at 6 months.

**Secondary outcomes:**
► Patient reported quality of life at 6 and 12 months, including SF-36 global and domain-specific scores, EQ-5D-5L summary index and visual analogue scale, and AFEQT overall score.
► Echocardiographic left-ventricular function at 12 months, including LVEF and diastolic function (E/e' and composite of diastolic indices).
► Functional assessment at 6 and 12 months, including 6 min walking distance and change in EHRA class.
► Change in NT-proBNP levels at 6 months.
► Change in heart rate from baseline and group comparison using 24 hours ambulatory ECG at end of uptitration.

**Feasibility assessment:**
► Successful methods for recruitment across primary and secondary care.
► Key issues that affect retention of participants, such as convenience, compliance and cross-over.
► Drug discontinuation rate and adverse reactions leading to drug discontinuation.
► Therapy-induced requirement for additional treatment (eg, pacemaker implantation).
► Population-specific SD and proportions to enable sample size calculation for a future trial.
► Assessment of unplanned hospital admissions and cardiovascular outcomes.

**Exploratory objectives:**
► Assessment of the validity and reproducibility of echocardiographic measures in patients with AF.
► Correlation of baseline measures, including quality of life questionnaires and unblinded baseline investigations such as quality of life, NT-proBNP, LVEF, E/e', EHRA class, intracellular biomarkers and heart rate.
► Impact of therapy on intracellular sodium and calcium concentration and cardiotonic steroid levels as biomarkers of cellular response.
► Impact of combination therapy on outcomes.
► Change in cognitive function at 12 months.
► Qualitative research of patient-reported quality of life using focus groups to explore patient acceptability, optimal delivery methods and responsiveness.
► Correlation of serum digoxin concentration with change in quality of life and intracellular methods.
► Cost-consequence economic analysis from an NHS healthcare perspective.

AF, atrial fibrillation; AFEQT, Atrial Fibrillation Effect on QualiTy of life questionnaire; NT-proBNP, N-terminal pro B-type natriuretic peptide; EHRA, European Heart Rhythm Association functional class; EQ-5D-5L, EuroQol five dimensions five level questionnaire; LVEF, left ventricular ejection fraction; NHS, National Health Service; SF-36, Short Form (36) Health Survey.

Serum will also be stored for the development of new blood-based and genetic biomarkers that aid in personalisation of rate control therapy.

**Statistical considerations**
The null hypothesis is of no difference in the physical functioning domain of the SF-36 quality of life questionnaire when comparing a strategy of digoxin versus beta-blocker therapy for initial rate control in older patients with permanent AF. The alternative hypothesis is superiority of one over the other therapy as an initial strategy of care. Randomising 144 patients we can assume an 85% power to detect an effect size of half a SD in a continuous outcome measure of quality of life (two-sided $\alpha$ of 0.05). Assuming that 10% of patients will be lost to follow-up, 160 patients are needed. There is some evidence from existing research to support the notion that the treatment effect could be this large. This includes a 17% improvement in SF-36 role-physical score in the rate control arm of the RACE study,[66] a 22% improvement in a proprietary symptom-checklist with CCB (compared with 8% change in those assigned beta-blockers),[19] and 17% improvement with rate control using SF-36 in the PIAF trial.[67] The RATE-AF trial will also help us to explore surrogates for clinical outcomes, such as LVEF using echocardiography

and BNP, and provide estimates for a future definitive trial of rate control in AF, including reliable information on recruitment rates, study drug titration, cross-over, retention and healthcare costs.

**Trial oversight, management and registration**
RATE-AF will be conducted in accordance with guidelines for Good Clinical Practice (GCP) and the Declaration of Helsinki, and has regulatory approval from the Medicines and Healthcare products Regulatory Agency.

Oversight will be provided by a Trial Steering Committee, comprising an independent Data Monitoring Committee and members of the RATE-AF Trial Management Group. This includes representatives of the patient and public involvement panel, involved in both the design and management of the trial. A Clinical Events Committee will be formed to adjudicate on adverse events.

The RATE-AF trial is registered at Clinicaltrials.gov (NCT02391337), ISRCTN (95259705) and EudraCT (2015-005043-13). Further information can be obtained from the trial website, http://www.birmingham.ac.uk/rate-af, and the trial protocol (see online supplementary material). The protocol was developed in accordance with the Standard Protocol Items for Randomized Trials

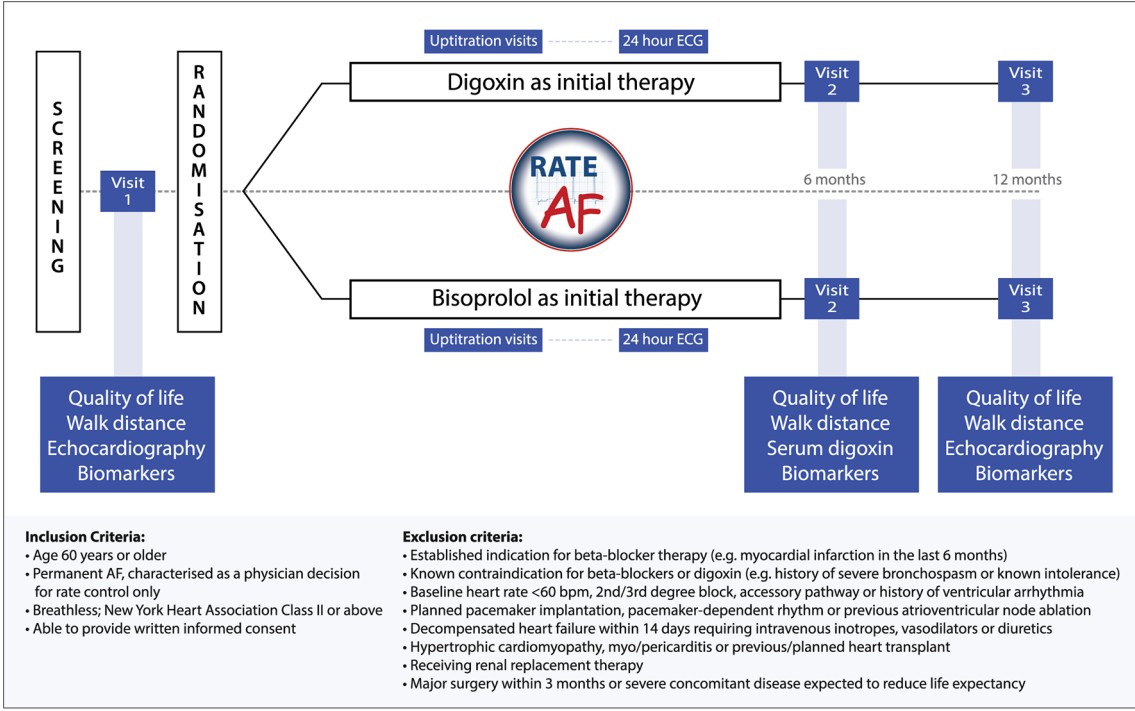

**Figure 3** The RAte control Therapy Evaluation in permanent Atrial Fibrillation (RATE-AF) trial schema. Trial flow chart, including major end points and inclusion/exclusion criteria.

**Table 1** Patient-reported quality of life questionnaires used in RAte control Therapy Evaluation in permanent Atrial Fibrillation

| Questionnaire | Details | Advantages and disadvantages |
|---|---|---|
| Short Form (36) Health Survey[80] | Generic instrument with 4-week recall period in eights domains (vitality, physical functioning, bodily pain, general health perceptions, physical role functioning, emotional role functioning, social role functioning and mental health) 11 subdivided questions, each recorded with a Likert scale Scoring: each response is given a numerical value (0–100, with 100 representing the best level of functioning possible), which are averaged across each domain | Extensively validated across a wide variety of conditions and the elderly[81] Not specific to AF and hence other comorbidities may dominate responses Requires a licence fee |
| EuroQol five dimensions five level questionnaire[82 83] | Generic instrument about today's health with a five-answer scale in five domains (mobility, self-care, usual activities, pain/discomfort and anxiety/depression) Scoring: each question is scored (1–5, with 1 representing the best health). The overall profile can be indexed to country-specific value sets giving a continuous value Also includes a visual analogue scale denoting current health perception (0–100 scale, with 100 representing the best health the patient can imagine) | Simple questionnaire that is quick to complete and includes a visual scale Extensive utilisation, particularly for heath economic assessment, with improvement discrimination over prior versions[84] Not specific to AF and hence other comorbidities may dominate responses |
| Atrial Fibrillation Effect on QualiTy of life questionnaire[85] | AF-specific quality of life instrument with 4-week recall period in domains relating to symptoms, daily activities and treatment 20 questions (18 on health-related quality and life and 2 on treatment satisfaction), each recorded with a 7-point Likert scale Scoring: responses to the 18 questions are summed and converted to a continuous score (0–100, with 100 corresponding to no patient concerns nor disability due to AF) Component domains are scored in a similar way | Specific to the impact of AF on quality of life Better than other AF-specific tools in a systematic review of methodological/psychometric assessment[68] Limited validation as yet in comparison with generic tools,[86 87] particularly for clinical responsiveness Licence fee may apply |

statement,[72] and the latest guidance from the International Society for Quality of Life Research Best Practice taskforce.[73–75]

## Ethics and dissemination

The trial has ethical approval from the East Midlands—Derby Research Ethics Committee (16/EM/0178) and approval from the National Health Service Health Research Authority (IRAS project ID: 191437).

The research findings will be submitted for publication to peer-reviewed journals after review by the oversight committees and the Patient Involvement Panel, and presented to relevant national and international meetings. Trial participants will be sent a lay summary of the final results of the trial, written by the Patient Involvement Panel.

## CONCLUSION

Defining appropriate rate control therapy is vital, particularly in the rapidly growing number of older patients with permanent AF, where current evidence is extremely limited. Rate control is an integral part of management in almost all patients with AF, but hardly any controlled trial evidence exists to guide the choice of agents. This is unacceptable in light of the potential benefits and possible adverse effects of treatment. In addition, the complete lack of data on the impact of medical therapy on symptom burden and heart function necessitate a programme of reproducibility and validity of both patient-reported quality of life and cardiac imaging in AF. The RATE-AF trial will answer key clinical questions about how to initiate therapy in order to improve patient well-being, stratified by relevant patient characteristics such as baseline symptoms, systolic and diastolic cardiac function, and biomarkers of treatment effect.

**Correction notice** This article has been corrected since it first published. The acronym for the Chief Investigator has been corrected. The Funding information has been updated with the correct ID of the Career Development Fellowship. Other typos and encoding errors have been corrected in the abstract, main text and reference list.

**Acknowledgements** We would like to acknowledge other members of the wider RAte control Therapy Evaluation in permanent Atrial Fibrillation team, including Karina Bunting, Patience Domingos, Dannie Fobian, Margaret Grant, Emma Hayes, Hannah Lack, Susan Jowett, Jonathan Mathers and Davor Pavlovic (University of Birmingham). We are indebted to the independent members of the trial oversight committees, as well as the Patient and Public Involvement Team.

**Contributors** The manuscript was drafted by DK who is the Chief Investigator for the RATE-AF trial. MG and GYHL are Principal Investigators. MC, PK, RPS and JNT are members of the Trial Management Group. JJD, SM and GS are representatives from the Clinical Trials Unit. MS is the Lead for the Patient Involvement Panel, and a member of the Steering Committee. All authors contributed to the writing of the RATE-AF protocol or patient information, and edited this manuscript for intellectual content.

**Funding** DK and the RATE-AF trial are supported by the National Institute of Health Research (NIHR) as part of a Career Development Fellowship (CDF-2015-08-074). The opinions expressed in this paper are those of the authors and do not represent the NIHR or the UK Department of Health.

**Competing interests** None of the authors report a conflict of interest. All authors have completed the ICMJE uniform disclosure form (www.icmje.org/coi_disclosure.pdf) and declare: DK reports grants from Menarini, during the conduct of the study; non-financial support from Daiichi Sankyo and personal fees from AtriCure, outside the submitted work. MC reports grants from the National Institute of Health Research, during the conduct of the study; and personal fees from Astella Pharma and Ferring Pharma, outside the submitted work. PK reports consulting fees and honoraria from Bayer Healthcare, Boehringer Ingelheim, Bristol-Myers Squibb, Daiichi Sankyo, Medtronic, Pfizer and Servier, all outside the submitted work; research grants from Bristol-Myers Squibb, Pfizer, Cardiovascular Therapeutics, Daiichi Sankyo, Sanofi, St. Jude Medical, German Federal Ministry for Education and Research (BMBF), Fondation Leducq, German Research Foundation (DFG), European Union, British Heart Foundation and Medical Research Council UK, all outside the submitted work; and is listed on two patent applications on AF therapy and markers for AF, both outside the submitted work. GYHL has served as a consultant for Bayer, Astellas, Merck, AstraZeneca, Sanofi, BMS/Pfizer, Biotronik, Portola and Boehringer Ingelheim, and has been on the speaker's bureau for Bayer, BMS/Pfizer, Boehringer Ingelheim and Sanofi Aventis. RPS is the President of the British Society of Echocardiography. JJD, MG, MS. JNT, SM, GS report no competing interests.

**Patient consent** Informed written consent was obtained from all trial participants using HRA and ethics-approved consent forms.

**Ethics approval** East Midlands-Derby Research Ethics Committee (16/EM/0178).

**Provenance and peer review** Not commissioned; externally peer reviewed.

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
