## [Reviewer comments · BMJ Open]

ARTICLE DETAILS

TITLE (PROVISIONAL)	A review of rate control in atrial fibrillation, and the rationale and protocol for the RATE-AF trial
AUTHORS	Kotecha, Dipak; Calvert, Melanie; Deeks, Jon; Griffith, Mike; Kirchhof, Paulus; Lip, Gregory; Mehta, Samir; Slinn, Gemma; Stanbury, Mary; Steeds, Richard; Townend, Jonathan

VERSION 1 - REVIEW

REVIEWER	Udo Bavendiek Hannover Medical School Dept. of Cardiology and Angiology Hannover, Germany
REVIEW RETURNED	20-Dec-2016

GENERAL COMMENTS	This is an important pilot-study for set-up of an urgently needed trial comparing beta-blocker and cardiac-glycosides for rate control in AF. Overall, the trial is nicely described and explained. The following points have to be clarified/corrected: Page 10, line 44: NYHA-classification should only be used for clinical presence of heart failure symptoms overall and not for grading of breathlessness only. This could cause misinterpretation like general assumption that breathlessness = heart failure, which is not true. Why not just NYHA-classification was used for inclusion criteria? The authors should explain. page 11, line 35 ff.: Why the authors used up titration to reach resting heart rate ≤ 100 bp, despite the lacking evidence of heart rate limits for morbidity/mortality in AF. Furthermore, digoxin-concentrations after up-titration are determined. The authors should explain and discuss these points in more detail in the paper according to the trial protocol. page 12, line 15: most likely "allow" is missing.
--

REVIEWER	Jonathan Piccini Duke University Medical Center, USA
REVIEW RETURNED	23-Dec-2016

GENERAL COMMENTS	Kotecha and colleagues have submitted a manuscript detailing the rationale and design of the RATE-AF trial. The study is an open-label pilot trial randomizing 160 subjects with NYHA class II heart failure to either treatment with bisoprolol or digoxin. The primary endpoint is the physical component summary of the SF-36. Key secondary outcomes include other quality of life measures, echocardiographic determined left ventricular function, 6-minute walk distance, change in heart rate, and brain natriuretic peptide
---

	levels at 6-months. The study will provide very useful data regarding rate control in AF and will help inform larger definitive trials focused on cardiovascular outcomes. Specific Comments: 1 – Very little of the manuscript is devoted to explaining the selection of the beta-blocker used in the study. The majority of existing data in AF is with metoprolol and carvedilol. Why was bisoprolol chosen? 2 – The stratification by EHRA functional class and sex is a nice feature, but will the EHRA score be as valuable since all patients will have NYHA II or greater dyspnea? Moreover, wouldn't HFpEF vs HFrEF be a more compelling stratification? 3 – Digoxin is a positive inotrope. Like many inotropes, it may improve symptoms but worsen mortality and other outcomes. The low power to detect a difference in hard outcomes is a significant risk of the small sample size of the study. 4 – The introduction and rationale section total 5 pages. This text could be shortened significantly without loss of meaningful content. 5 – The rationale section seems "one-sided" in many sections. For example, RACE II has several limitations, including low power, the inclusion of many endpoints not felt to be impacted by rate control (e.g. bleeding), and relatively good rate control in the "lenient" arm. Some of these limitations should be mentioned for balance. 6 – Similarly, the authors neglect to mention the body of data that suggests an association with harm in patients with AF treated with digoxin. More balance would improve the paper and make it more compelling. 7 – Why not make an objective measure of functional status (6 min walk) or heart failure severity (BNP) a co-primary endpoint?
--	---

REVIEWER	Cathy A. Jenkins Vanderbilt University Medical Center
REVIEW RETURNED	30-Jan-2017

GENERAL COMMENTS	This manuscript describes the protocol driving the RATE-AF trial. This trial attempts to better understand which rate control therapy for patients with permanent AF has better quality of life, ventricular output, and exercise capacity outcomes measured at 6 and/or 12 months. While this was well-planned, there are a few areas that need additional explanation: 1) In reading through the protocol and the manuscript, I was surprised to read in the manuscript that 'A key objective of the trial is to improve the methods used for measuring quality of life in AF patients'. This does not seem highlighted to me at all in reading through either. Is that true? And if so, can this be more fully explained? 2) When describing the stratified randomization process, it is stated that a 'minimisation algorithm' will be used. Does this mean some kind of adaptive trial design? Later, in the protocol, it describes what
--

	seems to be block randomization. Can you clarify? 3) Can you more fully describe the quality of life measure being used. You say it is continuous. What is its range? How is it determined? 4) While the sample size justification is included, no information is included as far as the analysis plan in the manuscript. A little more detail is included in the protocol; however, when discussing the regression, it only mentions 'minimisation variables'. Can this be further explained? And when it says 'all' of these variables, will over-fitting be an issue? If so, how will those to be included be determined? 5) Several times when discussing previous RCTs, rather strong language has been used to indicate null findings. For example, in discussing results of a meta-analysis looking at mortality and hospital admission in patients with HF, reduced EF, and concomitant AF, it is stated that 'all RCT data has shown that beta-blockers do not reduce all-cause mortality or hospital admissions'. This seems overly strong as absence of evidence does not indicate evidence of absence. Similarly, it is stated that 'Beta-blockers did not improve arrhythmia-related symptoms in an RCT of 60 low-risk patients with permanent AF'. Especially for trials of such small size, even with RCTs, at best the results failed to find an association rather than definitively proved as the current language suggests.
--	--

VERSION 1 – AUTHOR RESPONSE

Reviewer: 1

Reviewer Name: Udo Bavendiek

Institution and Country: Hannover Medical School, Dept. of Cardiology and Angiology, Hannover, Germany Please state any competing interests or state 'None declared': None declared.

Please leave your comments for the authors below

This is an important pilot-study for set-up of an urgently needed trial comparing beta-blocker and cardiac-glycosides for rate control in AF. Overall, the trial is nicely described and explained.

Thank you for these comments.

The following points have to be clarified/corrected:

Page 10, line 44:

NYHA-classification should only be used for clinical presence of heart failure symptoms overall and not for grading of breathlessness only. This could cause misinterpretation like general assumption that breathlessness = heart failure, which is not true. Why not just NYHA-classification was used for inclusion criteria? The authors should explain.

We agree with the reviewer that there can be confusion regarding the use of NYHA class in patients with AF. The NYHA classification system is widely used (in contrast to the more

recently proposed EHRA classification), and gives an assessment of the functional impact of breathlessness. Further, 'heart failure' is present in over 50% of patients with AF.

The distinction of whether dyspnoea in AF patients in the presence of preserved ejection fraction is regarded as heart failure or not remains controversial (see our recent article in JACC; Kotecha D, Lam CS, Van Veldhuisen DJ, et al.; Heart failure with preserved ejection fraction and atrial fibrillation - Vicious twins. J Am Coll Cardiol. 2016;68:2217-2228), particularly as BNP levels are consistently raised, and often in the heart failure range in AF patients.

Similarly, echo parameters have not been validated in AF for systolic function (Kotecha D, Popescu BA, Steeds RP, et al.; Is echocardiography valid and reproducible in patients with atrial fibrillation? A systematic review. Europace. 2017: in press). Whether mild systolic dysfunction truly represents heart failure or merely issues with accurately determining left-ventricular function in AF due to variable RR intervals is unknown.

We do not believe it is possible in this article to do this controversy justice, but hope that the RATE-AF trial will provide a unique perspective on some of these issues. We will be performing detailed and blinded assessment of left-ventricular systolic and diastolic function, correlation with BNP, and analysis of the EHRA vs NYHA classification systems.

We have added the following to the clinical practice section:

Exploratory work and clinical practice improvement:

Optimal acquisition of echocardiography in patients with AF will be determined by reproducibility studies, comparing repeated measures of systolic/diastolic function according to cardiac cycle length. ~~The aim of this work is to produce a standardised protocol of echocardiography in patients with AF. The RATE-AF trial will address the evidence-gaps we have identified in a systematic review of echocardiography in patients with AF⁶⁷, and explore the diagnostic difficulty of categorising heart failure in the context of AF (particularly with preserved ejection fraction, where symptom classification is confounded and BNP levels are consistently raised due to AF).~~

page 11, line 35 ff.: Why the authors used up titration to reach resting heart rate ≤ 100 bp, despite the lacking evidence of heart rate limits for morbidity/mortality in AF. Furthermore, digoxin-concentrations after up-titration are determined. The authors should explain and discuss these points in more detail in the paper according to the trial protocol.

Thank you for this important point. We agree that the relationship between heart rate and prognosis is unclear in patients with AF (with or without concomitant heart failure). The prevailing opinion from UK cardiologists (and indeed many around the world) is that reducing tachycardia will prevent development or deterioration in systolic and diastolic dysfunction (which, of course, has yet to be proven). Nonetheless, this principle is commonplace, and needs to be integrated into this pragmatic clinical trial. In the RATE-AF trial, we are comparing the effectiveness of digoxin versus beta-blockade for rate control. We are not testing the merits of heart rate reduction, although of course we will have indirect information, and the relative effects of these agents on heart rate.

We have added the following sentence to clarify (but have steered clear on any justification, as the RATE-AF trial will not test this hypothesis):

Study procedures and outcomes:

Participants will then receive study medication (bisoprolol 1.25-15 mg or digoxin 62.5-250 µg once daily), with scheduled uptitration visits to attain a heart rate at rest of ≤ 100 bpm.

This heart rate is in line with international recommendations¹ and was chosen pragmatically to reflect the opinion of many cardiologists that tachycardia can lead to, or worsen, systolic and diastolic dysfunction.

page 12, line 15: most likely "allow" is missing.

The section on Trial oversight has now been amended as per the Editor's request.

Reviewer: 2

Reviewer Name: Jonathan Piccini

Institution and Country: Duke University Medical Center, USA Please state any competing interests or state 'None declared': None declared.

Please leave your comments for the authors below

Kotecha and colleagues have submitted a manuscript detailing the rationale and design of the RATE-AF trial. The study is an open-label pilot trial randomizing 160 subjects with NYHA class II heart failure to either treatment with bisoprolol or digoxin. The primary endpoint is the physical component summary of the SF-36. Key secondary outcomes include other quality of life measures, echocardiographic determined left ventricular function, 6-minute walk distance, change in heart rate, and brain natriuretic peptide levels at 6-months. The study will provide very useful data regarding rate control in AF and will help inform larger definitive trials focused on cardiovascular outcomes.

Specific Comments:

1 – Very little of the manuscript is devoted to explaining the selection of the beta-blocker used in the study. The majority of existing data in AF is with metoprolol and carvedilol. Why was bisoprolol chosen?

Thank you for this point. Bisoprolol is by far the most common beta-blocker used by cardiologists in the UK, and has also largely replaced atenolol in general practice. Pharmacologically it has the advantage of a long half-life, rendering it suitable for long-term rate control using *once daily* dosing (as with digoxin). All clinicians are comfortable with its use in patients with AF (please bear in mind this study is open-label). In contrast, the use of metoprolol and carvedilol are mostly restricted to heart failure with reduced ejection fraction (and even then, are infrequently prescribed in the UK NHS).

2 – The stratification by EHRA functional class and sex is a nice feature, but will the EHRA score be as valuable since all patients will have NYHA II or greater dyspnea? Moreover, wouldn't HFpEF vs HFrEF be a more compelling stratification?

Stratification of randomisation is only designed to prevent major imbalance between treatment groups for specific factors that are known to influence treatment response. There is evidence for this in AF patients for gender (Schnabel RB, Pecun L, Johannsen SS, Ojeda FM, Lucerna M, Blankenberg S, Darius H, Kotecha D, De Caterina R, Kirchhof P; Gender differences in clinical presentation and predictors of one-year outcomes in atrial fibrillation. *Heart*. 2017; in press) and for EHRA class (Wynn GJ, Todd DM, Webber M, Bonnett L, McShane J, Kirchhof P, Gupta D; The European Heart Rhythm Association symptom classification for atrial fibrillation: validation and improvement through a simple modification. *Europace*. 2014;16:965-972). The EHRA score has specifically been designed to differentiate AF-related symptoms from other (e.g. heart failure related) symptoms. We agree that further validation of this score would be desirable. This can be expected as more researchers use the score.

HFpEF vs. HFrEF would not have been an appropriate stratification variable, as this would have required echocardiography as part of the screening process. Our aim was to make this

trial as external valid as possible, and so we have limited the screening and exclusion criteria as far as possible.

3 – Digoxin is a positive inotrope. Like many inotropes, it may improve symptoms but worsen mortality and other outcomes. The low power to detect a difference in hard outcomes is a significant risk of the small sample size of the study.

The trial is not designed or powered to look at clinical outcomes. The available randomised data (all in heart failure), suggests that low-dose digoxin does not increase mortality (Ziff OJ, Lane DA, et al.; Safety and efficacy of digoxin: systematic review and meta-analysis of observational and controlled trial data. BMJ. 2015;351:h4451). This is the first randomised trial of digoxin in patients actually in AF during therapy. We agree with this reviewer that the outcomes of this study may underpin the need for a further trial of clinical outcomes, should RATE-AF demonstrate safety and feasibility.

4 – The introduction and rationale section total 5 pages. This text could be shortened significantly without loss of meaningful content.

Thank you. We wanted to provide a detailed literature review of rate control therapy in AF. The paper combines both a narrative review and in addition, the rationale for our protocol, within a single open-access article. In response to this comment, we have gone through and reduced replication of content in all of these sections while maintaining the dual purpose of the paper.

5 – The rationale section seems “one-sided” in many sections. For example, RACE II has several limitations, including low power, the inclusion of many endpoints not felt to be impacted by rate control (e.g. bleeding), and relatively good rate control in the “lenient” arm. Some of these should limitations should be mentioned for balance.

Whilst we agree that RACE II has some limitations, the study team should be congratulated as this remains one of the highest-quality trials in AF rate control (and one of the only randomised studies). We have amended the following:

What is the optimal heart rate target in AF?

In the RACE II trial of 614 randomised patients with permanent AF, there were no benefits of strict (<80 bpm at rest) compared to lenient rate control (resting heart rate <110 bpm) over 3 years of follow-up.³⁴ Although interpretation was limited by the narrow difference in heart rate between groups, lenient rate control was found to be non-inferior with an adjusted hazard ratio of 0.80 (90% CI 0.55-1.17) for a wide composite and cumulative of adverse clinical outcomes (in- 12.9%, compared to 14.9% in the strict control arm).

6 – Similarly, the authors neglect to mention the body of data that suggests an association with harm in patients with AF treated with digoxin. More balance would improve the paper and make it more compelling.

The only data suggesting harm from digoxin in AF patients is from observational data and post-hoc analyses of randomized trials testing other interventions (e.g. oral anticoagulation). We have already shown that bias is directly related to the finding of adverse prognosis from digoxin (Ziff OJ, Lane DA, et al.; Safety and efficacy of digoxin: systematic review and meta-analysis of observational and controlled trial data. *BMJ*. 2015;351:h4451). It would not be appropriate to justify biased observational analyses, or suggest that such studies be considered as equivalent to randomised controlled trials; RATE-AF will be a first step to obtain the data needed to define the role of digoxin in contemporary management of AF patients.

We have added a qualification for our statement, but do not think this is the correct place to raise the complex issues relating to observational data analysis.

Do outcomes vary with different rate control therapies?

Similarly, ~~after accounting for the fact that sicker patients tend to receive digoxin more often,~~ the use of digoxin was not associated with any increase, or reduction, in mortality in a comprehensive systematic review.²¹ This finding deviates from prior observational analyses which are confounded by the fact that sicker patients tend to receive digoxin more often, which can only be addressed within a randomised trial.

7 – Why not make an objective measure of functional status (6 min walk) or heart failure severity (BNP) a co-primary endpoint?

Both 6-minute walk distance and BNP are specified secondary endpoints. In general, composite and co-primary endpoints are less desirable and the statistical issues with such approaches are poorly appreciated, particularly with regard to multiplicity and interpretation of divergent findings. Please note that the trial has undergone multiple review processes at the funding agency (National Institute of Health Research; NIHR) and regulators, and recruitment has already commenced with the listed outcomes.

Reviewer: 3

Reviewer Name: Cathy A. Jenkins

Institution and Country: Vanderbilt University Medical Center Please state any competing interests or state 'None declared': None declared

Please leave your comments for the authors below

This manuscript describes the protocol driving the RATE-AF trial. This trial attempts to better understand which rate control therapy for patients with permanent AF has better quality of life, ventricular output, and exercise capacity outcomes measured at 6 and/or 12 months. While this was well-planned, there are a few areas that need additional explanation:

1) In reading through the protocol and the manuscript, I was surprised to read in the manuscript that 'A key objective of the trial is to improve the methods used for measuring quality of life in AF patients'. This does not seem highlighted to me at all in reading through either. Is that true? And if so, can this be more fully explained?

Thank you for this important point. Indeed, there is great clinical need to find a method that can accurately measure quality of life and be useful for serial follow-up of patients. We have just published a systematic review on this issue, which confirms the AFEQT questionnaire as the most valid and reproducible questionnaire (hence our choice for this trial), but identified concerns about clinical and measurement validity that we will test during RATE-AF (Kotecha D, Ahmed A, Calvert M, et al.; Patient-Reported Outcomes for Quality of Life Assessment in Atrial Fibrillation: A Systematic Review of Measurement Properties. PLoS ONE. 2016;11:e0165790).

We have expanded on this area in the section of the manuscript titled 'Exploratory work and clinical practice improvement' (see below for edits), and in the protocol, these are detailed in Sections 2.5 and 9.5.2.

In brief, we will use qualitative methods (focus groups and interviews) which will follow a predefined schedule to address key concerns such as whether the AF and generic questionnaires are responsive to changes in patient well-being. A topic guide for the focus groups can be found at [this link](https://1drv.ms/b/s!AkdLtwkYAOEriqR3gbHKAHKFnPZyTA) <https://1drv.ms/b/s!AkdLtwkYAOEriqR3gbHKAHKFnPZyTA>, and will be included as an appendix to the publication on our qualitative research findings.

Exploratory work and clinical practice improvement

During the trial, qualitative research using focus groups and structured interviews will assess whether the quality of life questionnaires adequately and acceptably assess changes in symptom burden in a sample of patients from each treatment arm. We will also compare and contrast the generic and AF-specific questionnaires. The aim of this work is to identify the best processes improve the methods used for measuring patient-reported outcomes in AF, following on from a systematic review of measurement properties that identified key evidence-gaps and to address some of the limitations we have identified in published validation studies.⁶⁶

2) When describing the stratified randomization process, it is stated that a 'minimisation algorithm' will be used. Does this mean some kind of adaptive trial design? Later, in the protocol, it describes what seems to be block randomization. Can you clarify?

The randomisation employed is a standard RCT block approach, but which uses a minimisation algorithm that ensures good balance between groups for prognostic factors, even in small samples. With minimisation, the treatment allocated to the next participant enrolled in the trial depends on the characteristics of those participants already enrolled, such that each allocation minimises the imbalance across multiple factors. We have removed the word 'stratified' which may have caused confusion.

3) Can you more fully describe the quality of life measure being used. You say it is continuous. What is its range? How is it determined?

Thank you; we have amended the table to provide additional details (see below; due to limited space, we refer to the open-access article we have published on AF-specific quality of life: <http://journals.plos.org/plosone/article?id=10.1371/journal.pone.0165790>).

The AFEQT questionnaire used includes 20 questions, with domains on symptoms, daily activities, treatment concerns and treatment satisfaction. Answers are given on a 7-point Likert scale for each question. Scoring is performed in these domains and also globally. The generic questionnaires used are the SF-36 and EQ-5D-5L, both of which have established validation and scoring systems.

Table 3:

Questionnaire	Details	Advantages and disadvantages
SF-36 Short Form (36) Health Survey ⁷²	Generic instrument with 4-week recall period in eight domains (vitality, physical functioning, bodily pain, general health perceptions, physical role functioning, emotional role functioning, social role functioning and mental health). 11 subdivided questions, each scored recorded with a Likert scale. Scoring: Each response is given a numerical value (0 to 100, with 100 representing the best level of functioning possible), which are averaged across each domain.	Extensively validated across a wide variety of conditions and the elderly. ⁷³ Not specific to AF and hence other comorbidities may dominate responses. Requires a license fee.
EQ-5D-5L EuroQol five dimensions five level	Generic instrument about today's health with a five-answer scale in five domains (mobility, self-care, usual activities, pain/discomfort and anxiety/depression).	Simple questionnaire that is quick to complete and includes a visual scale. Extensive utilisation, particularly for health economic assessment, with

questionnaire ^{74 75}	Scoring: Each question is scored (1 to 5, with 1 representing the best health). The overall profile can be indexed to country specific value sets giving a continuous value. Also includes a visual analogue scale denoting current health perception (on a 0 to 100 scale, with 100 representing the best health the patient can imagine).	improvement discrimination over prior versions.⁷⁶ Not specific to AF and hence other comorbidities may dominate responses.
AFEQT Atrial Fibrillation Effect on Quality-of-life questionnaire ⁷⁷	AF-specific quality of life instrument with 4-week recall period in domains relating to symptoms, daily activities and concerns/satisfaction with current treatment. 20 questions (18 on health-related quality and life and 2 on treatment satisfaction), each scored recorded with a 7-point Likert scale. Scoring: Responses to the 18 questions are summed and converted to a continuous score (0 to 100, with 100 corresponding to no patient concerns nor disability due to AF). Component domains are scored in a similar way.	Specific to the impact of AF on quality of life. Better than other AF-specific tools using in a systematic review of methodological/psychometric assessment.⁶⁶ Limited validation as yet in comparison to generic tools^{78 79}, particularly for clinical responsiveness. License fee may apply.

4) While the sample size justification is included, no information is included as far as the analysis plan in the manuscript. A little more detail is included in the protocol; however, when discussing the regression, it only mentions 'minimisation variables'. Can this be further explained? And when it says 'all' of these variables, will over-fitting be an issue? If so, how will those to be included be determined?

The statistical analysis plan will be ratified by the Trial Steering Committee and independent Data Monitoring Committee before any blinded data is analysed. The minimisation variables discussed are gender and EHRA functional class at baseline, so we do not imagine that overfitting will be a problem. The statistical analysis plan will be included in the submission of the final results publication.

5) Several times when discussing previous RCTs, rather strong language has been used to indicate null findings. For example, in discussing results of a meta-analysis looking at mortality and hospital admission in patients with HF, reduced EF, and concomitant AF, it is stated that 'all RCT data has shown that beta-blockers do not reduce all-cause mortality or hospital admissions'. This seems overly strong as absence of evidence does not indicate evidence of absence. Similarly, it is stated that 'Beta-blockers did not improve arrhythmia-related symptoms in an RCT of 60 low-risk patients with permanent AF'. Especially for trials of such small size, even with RCTs, at best the results failed to find an association rather than definitively proved as the current language suggests.

Thank you – we have modified this language:

Do outcomes vary with different rate control therapies?

In patients with heart failure, reduced ejection fraction and concomitant AF, an individual patient level meta-analysis of ~~all-double-blind~~ RCT data has ~~shown-suggested~~ that beta-blockers do not reduce all-cause mortality or hospital admissions compared to placebo, in contrast to the substantial benefit seen in sinus rhythm.

Beta-blockers ~~did-not~~ were not associated with any improvement in arrhythmia-related symptoms in ~~an-a small~~ RCT of 60 low-risk patients with permanent AF, compared to diltiazem and verapamil which reduced the frequency of symptoms.

VERSION 2 – REVIEW

REVIEWER	Udo Bavendiek Dept. of Cardiology & Angiology, Hannover Medical School, Hannover, Germany
REVIEW RETURNED	14-Apr-2017

GENERAL COMMENTS	The authors nicely answered the points adressed by the reviewers. Only the uptitration-process in context of the digoxin-concentrations determined during the uptitration visits still has to be adressed in more detail. The authors should explain why no target range of digoxin-concentrations has been defined for digoxin dose-titration despite indications for adverse outcomes in the DIG-trial if digoxin serum-concentrations are above 1.0 ng/ml. Although the DIG-trial excluded patients with atrial fibrillation and at present there is no AF-trial adressing this properly this is an important point to be adressed. Furthermore, it should be explained how digoxin dose titration will be performed during the titration phase in the context of the determined digoxin concentrations as this is not explained in the trial protocol. If the authors adress this last point the manuscript should be accepted.
---

REVIEWER	Jonathan Piccini Duke University
REVIEW RETURNED	25-Mar-2017

GENERAL COMMENTS	I apologize, I uploaded my review a week ago but it appears that the upload was not received. To review, I appreciate the careful and thoughtful comments from the authors. I remain concerned over the potential for adverse outcomes with digoxin (and thus impairment of the interpretability of the results), particularly given the data from incidence use from the ARISTOTLE study. However, this study will provide much needed data in the field. There are not enough studies of rate control available to guide treatment of this common problem in cardiology and internal medicine.
---

REVIEWER	Cathy A. Jenkins Vanderbilt University Medical Center United States
REVIEW RETURNED	05-Apr-2017

GENERAL COMMENTS	The authors addressed all of my concerns adequately. The only one I had some further issue with was the one about the analysis plan not being fully explained but upon further review and their responses, I am satisfied.
--

VERSION 2 – AUTHOR RESPONSE

Reviewer: 2

Reviewer Name: Jonathan Piccini

Institution and Country: Duke University Please state any competing interests or state 'None declared': None

I apologize, I uploaded my review a week ago but it appears that the upload was not received. To review, I appreciate the careful and thoughtful comments from the authors. I remain concerned over the potential for adverse outcomes with digoxin (and thus impairment of the interpretability of the results), particularly given the data from incidence use from the ARISTOTLE study. However, this study will provide much needed data in the field. There are not enough studies of rate control available to guide treatment of this common problem in cardiology and internal medicine.

Response: Thank you for your detailed review.

Reviewer: 3

Reviewer Name: Cathy A. Jenkins

Institution and Country: Vanderbilt University Medical Center, United States Please state any competing interests or state 'None declared': None declared

The authors addressed all of my concerns adequately. The only one I had some further issue with was the one about the analysis plan not being fully explained but upon further review and their responses, I am satisfied.

Response: Thank you for your detailed review.

Reviewer: 1

Reviewer Name: Udo Bavendiek

Institution and Country: Dept. of Cardiology & Angiology, Hannover Medical School, Hannover, Germany Please state any competing interests or state 'None declared': None declared.

The authors nicely answered the points addressed by the reviewers.

Only the uptitration-process in context of the digoxin-concentrations determined during the uptitration visits still has to be addressed in more detail.

The authors should explain why no target range of digoxin-concentrations has been defined for digoxin dose-titration despite indications for adverse outcomes in the DIG-trial if digoxin serum-concentrations are above 1.0 ng/ml. Although the DIG-trial excluded patients with atrial fibrillation and at present there is no AF-trial addressing this properly this is an important point to be addressed. Furthermore, it should be explained how digoxin dose titration will be performed during the titration phase in the context of the determined digoxin concentrations as this is not explained in the trial protocol.

If the authors address this last point the manuscript should be accepted.

Response: Thank you. Unfortunately serum digoxin concentration (SDC) is a poor indicator of clinical response – we have published a paper on this issue (Ziff & Kotecha: Trends Cardiovasc Med. 2016;26:585-595) which reviews the data that digoxin toxicity can occur in the context of normal SDC, and conversely high SDC does not correspond to clinical toxicity. Currently, SDC is measured with an immunoassay, which may cross-react with other targets, including *endogenous* cardiotoxic steroids (CTS). These CTS are varied and present at different levels in different comorbidities (particularly in heart failure) and are likely to be a major confounder in the SDC immunoassay which is not sensitive enough to isolate specific CTS like digoxin. As part of the RATE-AF trial, we have secured funding to develop new methods of assessing digoxin, including cellular immunofluorescence models of sodium and calcium transport, and ultra-sensitive liquid chromatography-tandem mass spectrometry (LC-MS/MS).

Below are some references on this issue* – we have added the most recent (Dasgupta 2012) to the reference list of the article.

As this is a pragmatic clinical trial, clearly if SDC levels at 6 months (or during the uptitration schedules as clinically needed) are in toxic ranges or there is evidence of clinical toxicity, then digoxin dose will be reduced appropriately. However, there are few data to suggest that SDC is practically-helpful to determine digoxin dose. Rather, we instead will use low-dose digoxin therapy in all patients (the safety of which was demonstrated in the DIG randomised trial in sinus rhythm), which maintains the pharmacological properties of digoxin, whilst minimising adverse effects.

The following amendment has been made:

Blood samples from participants will analysed for the cellular effects of rate control, including intracellular sodium, calcium and endogenous cardiotoxic steroids (CTS) using photometry in cultured human cardiomyocytes. This work will give mechanistic insight into the cellular response to beta-blockers and digoxin, identify novel markers of treatment effect, and develop assays that are more robust than serum digoxin concentration (SDC) for determining individual patient dosage. SDC is an immunoassay known to be a poor marker of digoxin toxicity⁷⁰, which can cross-react with other targets⁷¹ (for example, endogenous CTS). Although SDC will be performed at six months follow-up and as required during the trial to advise clinicians on dose and avoid high digoxin levels, digoxin toxicity remains a clinical diagnosis at present. Serum will also be stored for the development of new blood-based and genetic biomarkers that aid in personalisation of rate control therapy.

***Reference list for reviewer:**

DASGUPTA, A. (2002) Endogenous and exogenous digoxin-like immunoreactive substances - Impact on therapeutic drug monitoring of digoxin. American Journal of Clinical Pathology, 118, 132-140.

GERS, N., JONES, T. & RG, M. (2010). Frequently discordant results from therapeutic drug monitoring for digoxin: clinical confusion for the prescriber. Internal Medical Journal, 40, 52-6.

DASGUPTA, A. (2012) Impact of Interferences Including Metabolite Crossreactivity on Therapeutic Drug Monitoring Results. Therapeutic Drug Monitoring, 34, 496-506.

VERSION 3 – REVIEW

REVIEWER	Udo Bavendiek Dept. of Cardiology & Angiology, Hannover Medical School, Hannover, Germany
REVIEW RETURNED	22-May-2017

GENERAL COMMENTS	I am satisfied with the authors' responses addressing the remaining point regarding digoxin-dose-titration and serum concentrations.
--